# Numerical Approach and Verification Method for Improving the Sensitivity of Ferrous Particle Sensors with a Permanent Magnet

**DOI:** 10.3390/s23125381

**Published:** 2023-06-06

**Authors:** Sung-Ho Hong

**Affiliations:** Department of Mechanical System Engineering, School of Creative Convergence Engineering, Dongguk University—WISE Campus, Gyeongju 38066, Republic of Korea; hongsh@dongguk.ac.kr; Tel.: +82-54-770-2211

**Keywords:** ferrous particle, permanent magnet, oil sensor, sensitivity

## Abstract

This study aimed to improve the sensitivity of ferrous particle sensors used in various mechanical systems such as engines to detect abnormalities by measuring the number of ferrous wear particles generated by metal-to-metal contact. Existing sensors collect ferrous particles using a permanent magnet. However, their ability to detect abnormalities is limited because they only measure the number of ferrous particles collected on the top of the sensor. This study provides a design strategy to boost the sensitivity of an existing sensor using a multi-physics analysis method, and a practical numerical method was recommended to assess the sensitivity of the enhanced sensor. The sensor’s maximum magnetic flux density was increased by around 210% compared to the original sensor by changing the core’s form. In addition, in the numerical evaluation of the sensitivity of the sensor, the suggested sensor model has improved sensitivity. This study is important because it offers a numerical model and verification technique that may be used to enhance the functionality of a ferrous particle sensor that uses a permanent magnet.

## 1. Introduction

Machine condition monitoring is a field that aims to improve the reliability of machines by detecting faults or failures through the use of sensors and measuring devices. By collecting and analyzing data and information, machine condition monitoring can prevent machine failure and enhance the maintenance level of a mechanical system [1,2,3]. Machine condition monitoring with oil analysis has been applied in various industrial fields such as automobiles, construction equipment, and power plants. Among various methods for analyzing lubricants, on-line methods using lubricant oil sensors are preferred. The reason for this is because these methods have advantages such as a reduction in human error, the prevention of major failure in the initial state, and low maintenance costs without needing professional skills [3,4].

The occurrence of wear is intricately connected to the condition, maintenance, and durability of machines. Therefore, it is crucial to determine the types or amount of wear [5]. It is also essential to identify particles that arise from wear within lubrication systems. These particles usually contain ferrous, non-ferrous, and non-metal debris such as ceramics and polymers. To measure wear particles, wear particle sensors commonly utilize inductance- and capacitance-based methods [6,7,8,9,10,11,12,13,14,15,16,17,18,19], acoustic methods based on ultrasonic transducers [20,21,22], optical methods [23,24,25,26], magnetic methods [27,28], and a method based on a combination of a permanent magnet and inductance [29]. Among them, ferrous particle sensors are widely used to diagnose machine condition because machines are made of iron as their main component.

Figure 1 shows a schematic of a ferrous particle sensor with a permanent magnet. The sensor consists of two units, each containing a sensing coil with inductance and a permanent magnet to attract ferrous debris at the tip. This sensor is capable of distinguishing between fine and coarse debris. It can prevent abrasive wear and damage because of the accumulation of ferrous particles of the permanent magnet [30]. Due to these advantages, its utilization is increasing among ferrous particle sensors.

While there have been many studies on the sensitivity of wearable particle sensors during development, there have been relatively few studies on the sensitivity of ferrous particle sensors [31,32,33,34,35,36]. Improvements in the sensor’s signal processing, modifications to its internal design, and other techniques may be used to increase the sensitivity of the ferrous particle sensor. The ferrous particle sensor has a cylindrical permanent magnet shape inside. As a result, ferrous particles are gathered not only on the sensor’s top but also on its sides. Thus, it is difficult to reflect the exact change in the number of ferrous particles attached to the sensor. This study focused on improving the sensitivity of the sensor by changing the design inside the sensor. This procedure involved creating a numerical model for the sensor in order to change its design effectively and presenting a numerical verification model to assess how sensitive the modification was. In addition, a new design model that improves the sensitivity of the existing sensor was presented, and the improvement was shown through an effective numerical verification model in terms of cost and time.

## 2. Numerical Model and Analysis

The numerical analysis consisted largely of an analysis of the change in magnetic flux density due to the shape of the sensor’s core and an analysis evaluating the effect of collecting ferrous particles in the flow. Through improvement of the performance of the sensor, it is intended to numerically show whether the sensitivity is improved by improving the collection effect of the sensor in the flow. Because this sensor measures the number of ferrous particles after attaching ferrous particles to the sensor with a permanent magnet, the collective effect of the sensor is important.

### 2.1. Analysis of Magnetic Flux Density for Changes in the Sensor’s Internal Design Parameters

First of all, a numerical model was developed to analyze the performance of the existing ferrous particle sensor. Figure 2 shows the numerical model of the ferrous particle sensor with a permanent magnet. It is hard to obtain the exact design parameters, including the sensor’s materials. Thus, the primary design parameters were roughly estimated by dismantling the sensor. These approximations were used in the analysis. Additionally, the magnetic core was made of low-carbon steel M-50. The magnetic flux density (B)–magnetic field intensity (H) curve was derived from data provided in the analysis program, as shown in Figure 3. The number of turns of the coil was 300, and the applied current was 0.1 A.

Figure 4 shows the mesh used in the numerical model of the sensor. In the analysis, triangle and quad meshes were mixed. The total number of meshes was 158,525. The meshes were densely applied around the sensor to increase the accuracy of the analysis. 

Figure 5 shows four sensor models in which the shape of the core inside the sensor is changed. The A-model is similar to the existing sensor model. In the B-model, the shape of the core was obliquely inclined to concentrate the magnetic flux density in the center. In the C-model, the magnetic flux density was reduced to the side of the sensor to focus the magnetic flux density on the upper part compared to the B-model. In the D-model, the area of the core below the coil was removed and the shape was changed to further improve the magnetic flux density on the upper part of the sensor. Geometries for the four models of the sensor are shown in Table 1.

This study utilized a multi-physics analysis method. COMSOL 6.0, commercial multi-physics software was used for numerical calculations. The sensor’s magnetic field was calculated based on Maxwell’s equations. The formulations used for calculating the magnetic field are shown in Equations (1)–(3): (1)∇×H=J
(2)$$J=\sigma (E+v\times B)+J_{e}$$
(3)B=∇×D
where *H*, *J*, *B*, and *D* are the magnetic field intensity [A/m], current density [A/m^2^], magnetic flux density [T], and magnetic vector potential [A], respectively. The force on a charge *σ* [C] moving with velocity *v* [m/s] in the presence of an electric and magnetic field *E* [V/m], *B*, is called the Lorentz force, and *J_e_* [A/m^2^] is an externally generated current density. In this program, the AC/DC module contains the electromagnetic field interface model, which calculates the magnetic flux of the ferrous particle sensor. For cases where currents and electromagnetic fields vary slowly, the induced displacement current can be ignored. This assumption, referred to as quasistatic approximation, is widely used in modeling low-frequency electromagnetic fields where the dimension of the structure is small compared to the wavelength ([37], p. 84). 

The magnetic flux density distributions for the four sensor models are shown in Figure 6. The flux density obtained from the simulation in Figure 6 is for the whole magnet. The B-model had a higher maximum magnetic flux density than the A-model. However, some magnetic flux density was generated at the side part (red dotted area) of the sensor. Thus, the sensitivity of the sensor was not greatly improved. Compared to the B-model, the C-model removed the area of the core below the coil. As a result, the magnetic flux density was concentrated in the upper part, and less magnetic flux density was generated in the side part. Therefore, the D-model improved the sensitivity of the sensor by changing the shape of the core. The D-model increased the maximum magnetic flux density by about 210% compared to the conventional sensor type (A-model) and improved the sensitivity by lowering the magnetic flux density of the side part.

### 2.2. Evaluating the Sensitivity of the Sensor in the Flow Field

In Section 2.1, the magnetic flux density of the sensor was improved. Although it is essential to fabricate an actual sensor and conduct an experiment for evaluating the performance, a method for verifying whether the sensitivity of the sensor has been improved is suggested using a numerical method. Evaluating the sensitivity of the sensor numerically is economical in terms of cost and time. However, verification through experiments for test devices and lubrication systems is absolutely necessary. This study focused on how to evaluate the sensitivity of the sensor with a numerical method. Among the four models of the sensor, only the A-model and D-model were evaluated for sensitivity numerically. This is because the A-model is a case in which the existing sensor is described, and the D-model is a case in which the magnetic flux density of the sensor is greatly improved.

The numerical analysis employed Navier–Stokes equations, the electromagnetic field interface model, and the particle tracing module. The AC/DC module contains the electromagnetic field interface model (Equations (1)–(3)), which calculates the magnetic flux of the ferrous particle sensor. The Navier–Stokes equations for the rotating domains are shown below:(4)∇⋅(ρv)=0
(5)ρ(v⋅∇)v+2ρΩ×v=∇⋅[−pI+τ]+F−ρ(Ω×(Ω×r))
where Ω, *I*, *τ*, and *F* mean the angular velocity [1/s], identity matrix, shear stress [N/m^2^], and volume force [N/m^3^], respectively.

The particle tracing module is utilized to compute individual particles’ paths by solving their equations of motion over time, which allows for the evaluation of discrete trajectories:(6)$$\begin{array}{c} \frac{d}{dt}(m_{p}v_{1})=F_{D}+F_{ext}\\ F_{D}=\frac{1}{\tau_{p}}m_{p}(v-v_{1}),\;\;\;F_{ext}=2\pi r_{p}^{3}\mu_{o}\mu_{r}K{\left|H\right|}^{2},\;\;\;\tau_{p}=\frac{\rho_{p}d_{p}^{2}}{18\mu } \end{array}$$
where *m_p_*, *v*_1_, *F_D_*, and *F_ext_* are the particle mass [kg], velocity vector of the particle [m/s], drag force [N], and magnetophoretic force [N], respectively. In addition, *τ_p_*, *d_p_*, *r_p_*, *ρ_p_*, *μ*, *μ_0_*, *μ_r_*, and *K* mean the particle velocity response time [s], particle diameter [m], particle radius [m], particle density [kg/m^3^], the viscosity of fluid [Pa∙s], vacuum permeability [kg∙m∙s/A^2^], relative permeability, and non-dimensional parameter, respectively ([37], pp. 272–304). 

The motion of particles in a fluid follows Newton’s second law, which states that the net force on an object is equal to the time derivative of its linear momentum in an inertial reference frame as shown in Equation (6). 

This numerical analysis model was proposed simply and uncomplicatedly to numerically evaluate the sensitivity of the improved ferrous particle sensor, not to analyze a specific mechanical system. This analysis was conducted using a planar symmetric model as shown in Figure 7. Various meshes such as tetrahedral, pyramid, prism, and hexahedral were used. The total number of elements was 872,406 in the A-model and 1,158,494 in the D-model. Moreover, a dense mesh was applied around the sensor to ensure analytical accuracy. The size of the flow channel was 140 mm (x) × 80 mm (y) × 80 mm (z). The flow was laminar. The working conditions for the numerical calculations are shown in Table 2. The particles used in the analysis were spherical and the material was iron with a density of 8030 kg/m^3^. During the initial five seconds of calculation, 1500 particles were injected from the particle injection area (blue shade area) at intervals of 0.5 s. The total number of injected particles was 15,000. The density and viscosity of the lubricant used in the analysis were 870 kg/m^3^ and 0.04 Pa∙s, respectively. Actual wear particles include non-magnetic particles and also consist of very diverse types of particles. This analysis is limited by the fact that the total number of particles can be less than 1% and that the shape of the particles can only be described as spherical.

In this analysis, not only the trajectories of the particles but also the number of particles collected by the sensor were evaluated. To check the number of particles attached to the sensor, a function called “particle counter” of the particle tracing module was used. The number of particles collected by the sensor was evaluated while the fluid velocity changed from 0.002 m/s to 0.1 m/s. The velocity of the flow in the analysis is small, so it is different from the actual hydraulic fluid. When the velocity of the fluid is high, the number of particles attached to the ferrous particle sensor decreases due to the inertial force of the fluid. In addition, when the speed of the fluid increases, the influence of the positioning of the sensor increases. Therefore, in order to easily compare the sensitivity improvement of the sensor, an analysis was performed under the condition of low speed. Since this sensor uses a permanent magnet to collect ferrous particles and then measures the number of ferrous wear particles through a change in the magnetic field, the sensor’s collecting effect is the most important factor of the sensor’s sensitivity. Therefore, the sensitivity of the sensor was evaluated by determining how many ferrous particles adhered to the sensor under several flow conditions.

Figure 8 shows particle trajectories with time in the A-model and D-model when the velocity of the fluid is 0.002 m/s. A total of 15,000 ferrous particles flowed from the inlet side to the outlet, and among them, some particles started to attach to the top of the sensor from about 30 s when the A-model was applied. In the case of the D-model, some particles began to adhere to the top of the sensor at a time similar to that of the A-model. However, the number of particles attached to the sensor of the D-model was greater than that attached to the sensor of the A-model.

Figure 9 shows the collected particles on the top of the sensor in the A-model and D-model when the fluid velocity is 0.002 m/s. As shown in Figure 9, the sensitivity of the sensor was evaluated by the number of particles collected on the top of the sensor. In Figure 9, the color of the particles indicates the magnitude of the magnetophoretic force. Particles collected in the central part of the top surface of the sensor are subjected to a large magnetophoretic force. The magnetophoretic forces acting on the ferrous particles are defined as *F_ext_* in Equation (6).

Compared to the existing sensor model (A-model), the improved sensor (D-model) investigated how the particle collecting effect appeared according to the change in fluid velocity as shown in Figure 10. When the fluid velocity was 0.002 m/s, the number of particles attached to the sensor was 3313 in the A-model and 3470 in the D-model. That is, the case of the D-model increased by about 4.7% compared to the case of the A-model in terms of the number of attached particles. When the fluid velocity increased to 0.02 m/s and 0.04 m/s, the results of the D-model increased by about 9.2% and 44%, respectively, compared to those of the A-model in terms of the number of attached particles. When the fluid velocity was 0.1 m/s, the D-model had 22 particles attached to the sensor, but the A-model had no particles attached to the sensor. It is difficult to attach particles to the sensor under conditions where the velocity of the fluid is higher. This is because the inertia force of the fluid increases as the velocity of the fluid increases. This result confirms that the sensor of the improved model shows a distinct improvement in sensitivity in a situation where the fluid velocity increases. In addition, the ratio of particles collected by the sensor will change according to the number of injected particles. The reason is that the magnetic flux density of the sensor is constant, and the number of particles directed around the sensor decreases when the number of injected particles decreases. Moreover, when pressure or temperature is a major factor in the analysis conditions, sufficient analysis is possible as a boundary condition. To take into consideration the change in temperature, the energy equation is additionally employed in the governing equations.

Figure 11 shows the magnetic flux density and magnetic force lines around the sensor in a fluid field where the fluid velocity is 0.1 m/s. The maximum magnetic flux density of the A-model was 0.436 T. However, the maximum magnetic flux density of the D-model was 0.913 T, which was about 209 % higher than that of the A-model. In addition, it can be seen that it is more advantageous to collect particles in the sensor because it is formed toward the upper end of the core in the D-model than in the A-model through lines of magnetic force around the core of the sensor. Therefore, a method for evaluating the sensitivity of the ferrous particle sensor with a permanent magnet using a numerical analysis based on multi-physics was proposed. Through this method, it was shown that the sensitivity of the design-changed model was improved in the flow field.

## 3. Conclusions

This study suggested a sensor model that could improve the sensitivity of an oil sensor by measuring the number of ferrous particles in lubricating oil. It also presented a numerical model for evaluating the sensitivity of the sensor in a flow field for the first time. The ferrous particle sensor attaches ferrous particles to the sensor using a permanent magnet and measures the number of attached ferrous particles. Moreover, it is widely used to diagnose the condition of lubrication systems such as engines and gearboxes. Since the sensitivity of this sensor is predominantly affected by particles collected on the top of the sensor, we tried to improve the sensitivity of the sensor by changing the shape of the core inside the sensor. Based on the analysis results of several models that changed the shape of the core, an improved model that reduced the magnetic flux density on the side of the sensor and increased the magnetic flux density on the top of the sensor was suggested. The maximum magnetic flux density of the improved model increased by about 210% compared to that of the existing model. A numerical method was suggested to evaluate the sensitivity of the sensor in the flow field. This numerical approach method is economical in terms of time and cost. Through sensitivity evaluation of the existing model and the proposed model under various fluid velocity conditions, it was confirmed that the proposed model improved sensitivity even in the flow field. However, the optimal sensor position considering the flow must be selected first, in order to verify the sensitivity of the sensor in the lubrication system where complex flow actually occurs. Moreover, a verification experiment must be performed for the test device in order to apply this sensor. Therefore, additional research related to this should be conducted in the near future. 

## Figures and Tables

**Figure 1 sensors-23-05381-f001:**
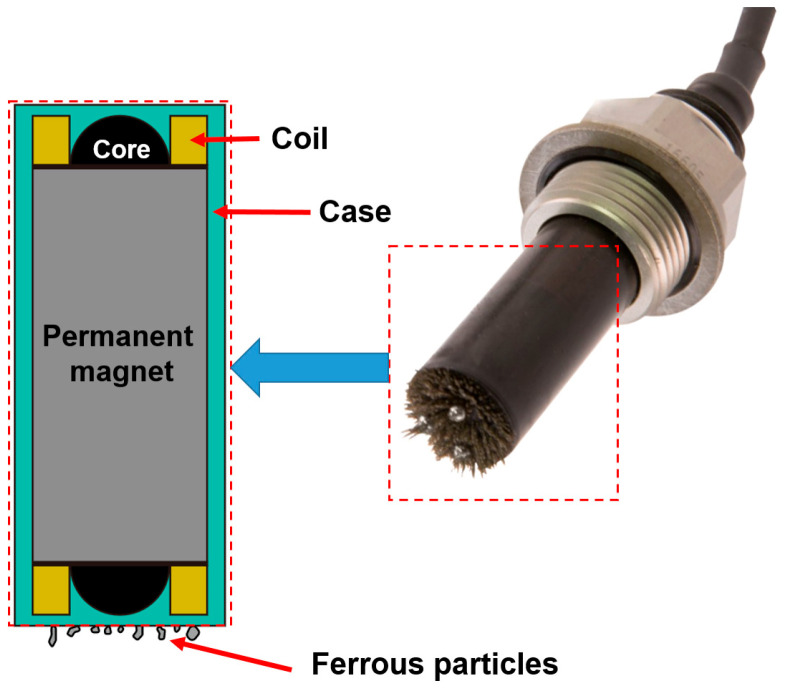
Schematic of the ferrous particle sensor with a permanent magnetic manufactured by Gill sensors.

**Figure 2 sensors-23-05381-f002:**
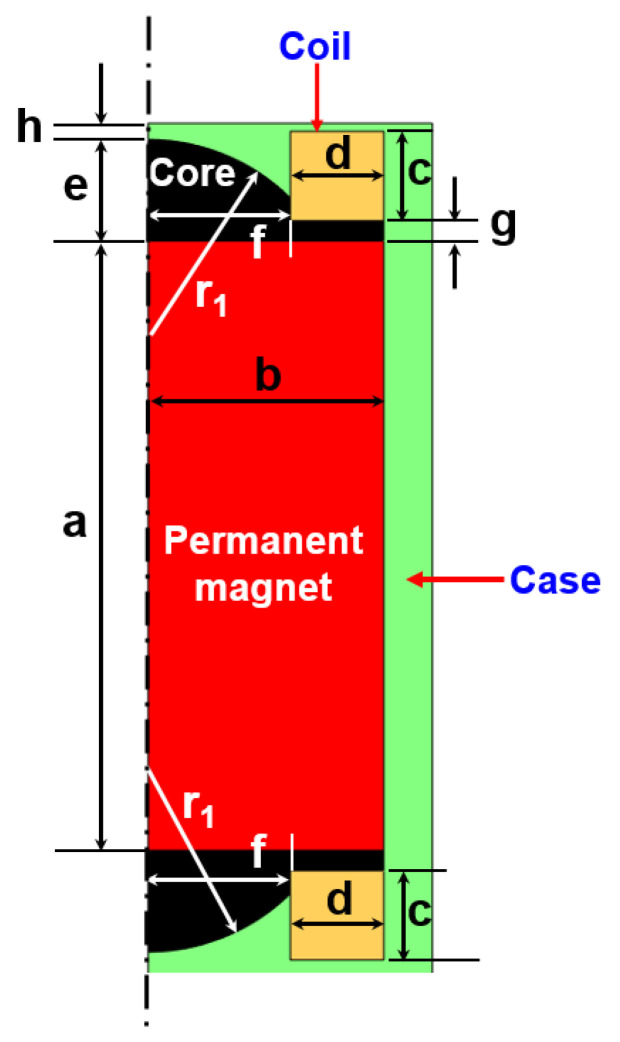
Axisymmetric section of the numerical model for the ferrous particle sensor.

**Figure 3 sensors-23-05381-f003:**
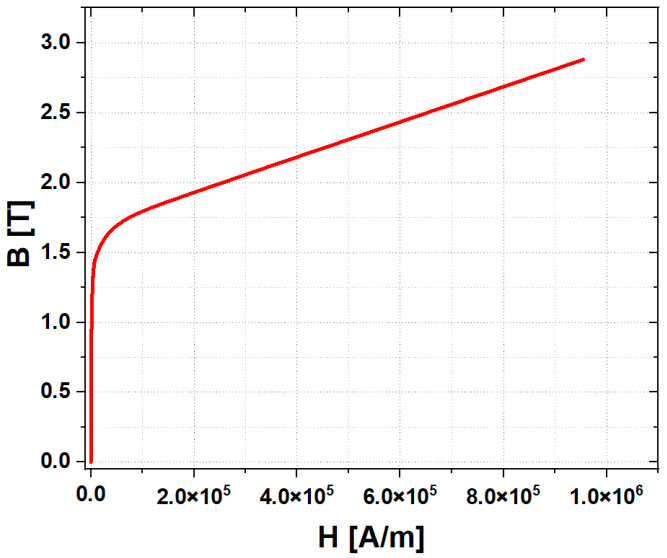
B-H curve of low-carbon steel M-50.

**Figure 4 sensors-23-05381-f004:**
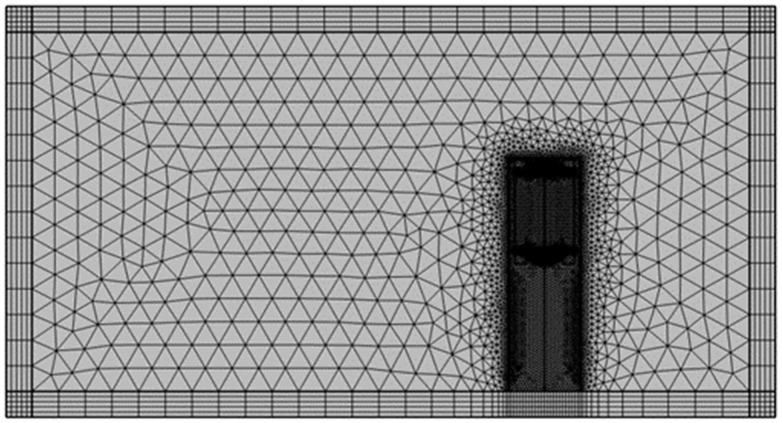
Meshes around the sensor.

**Figure 5 sensors-23-05381-f005:**
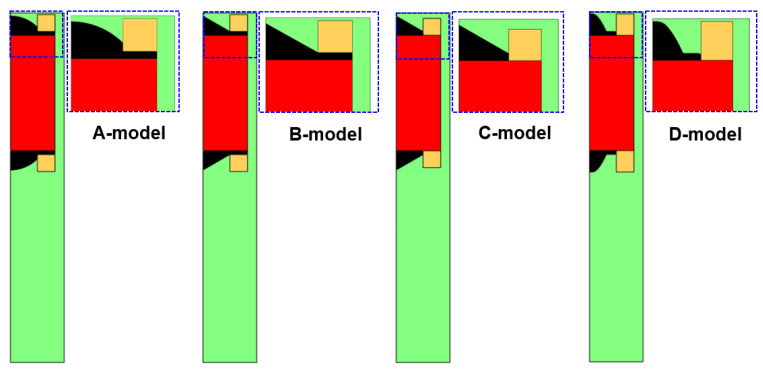
Four sensor models with different core shapes.

**Figure 6 sensors-23-05381-f006:**
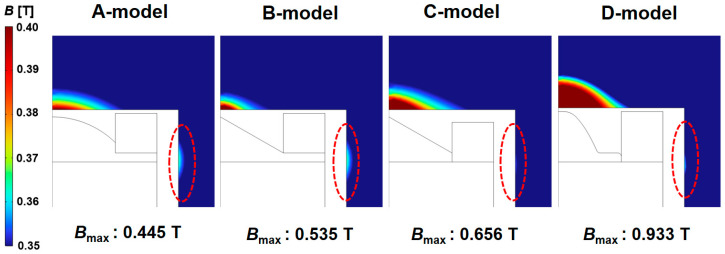
Magnetic flux density distributions for the four sensor models.

**Figure 7 sensors-23-05381-f007:**
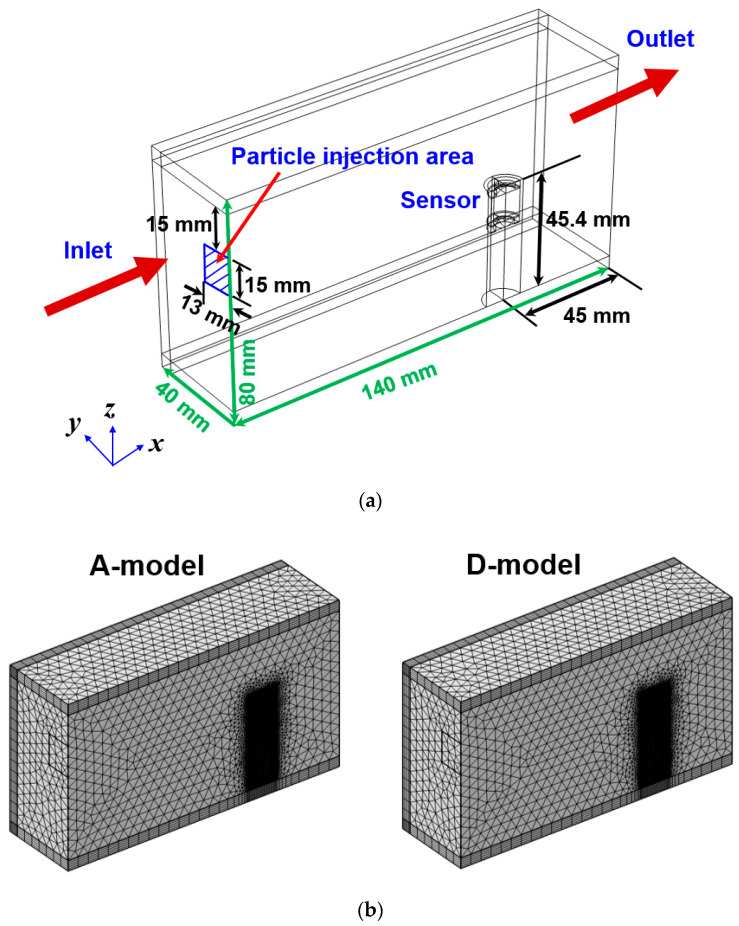
Numerical model used for sensitivity evaluation of the sensor: (**a**) control volume and (**b**) meshes of the A-model and D-model.

**Figure 8 sensors-23-05381-f008:**
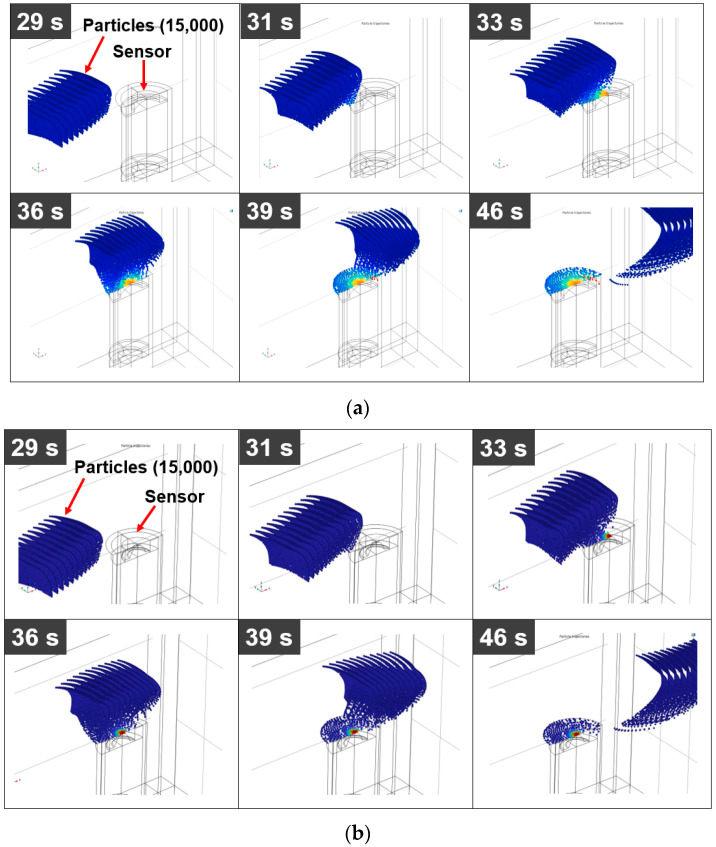
Particle trajectories with time: (**a**) A-model, *v* = 0.002 m/s and (**b**) D-model, *v* = 0.002 m/s.

**Figure 9 sensors-23-05381-f009:**
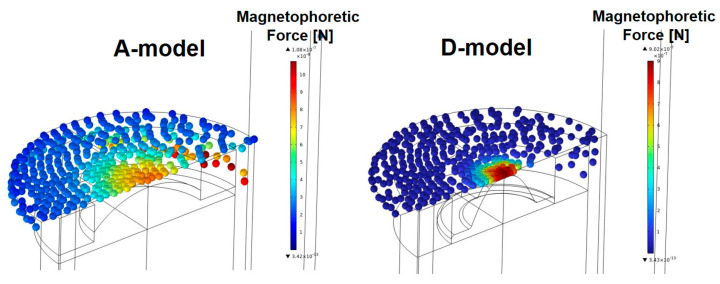
Number of particles collected on the top of the sensor in the A-model and D-model (*v* = 0.002 m/s).

**Figure 10 sensors-23-05381-f010:**
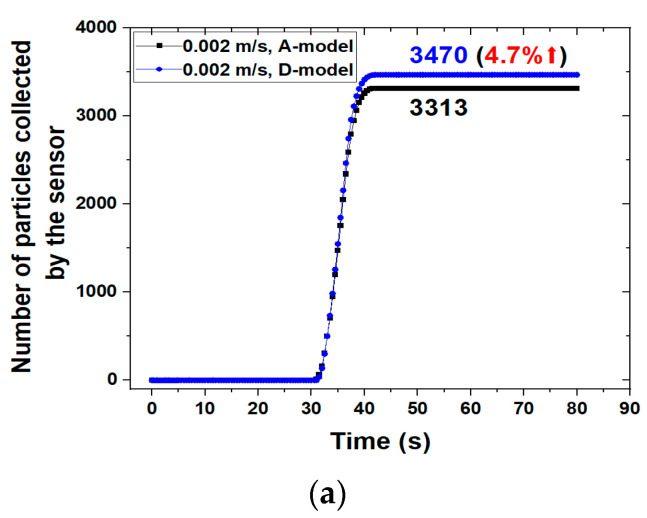
Number of particles collected by the sensor with variation in fluid velocity in the A-model and D-model: (**a**) *v* = 0.002 m/s; (**b**) *v* = 0.02 m/s; (**c**) *v* = 0.04 m/s; (**d**) *v* = 0.1 m/s.

**Figure 11 sensors-23-05381-f011:**
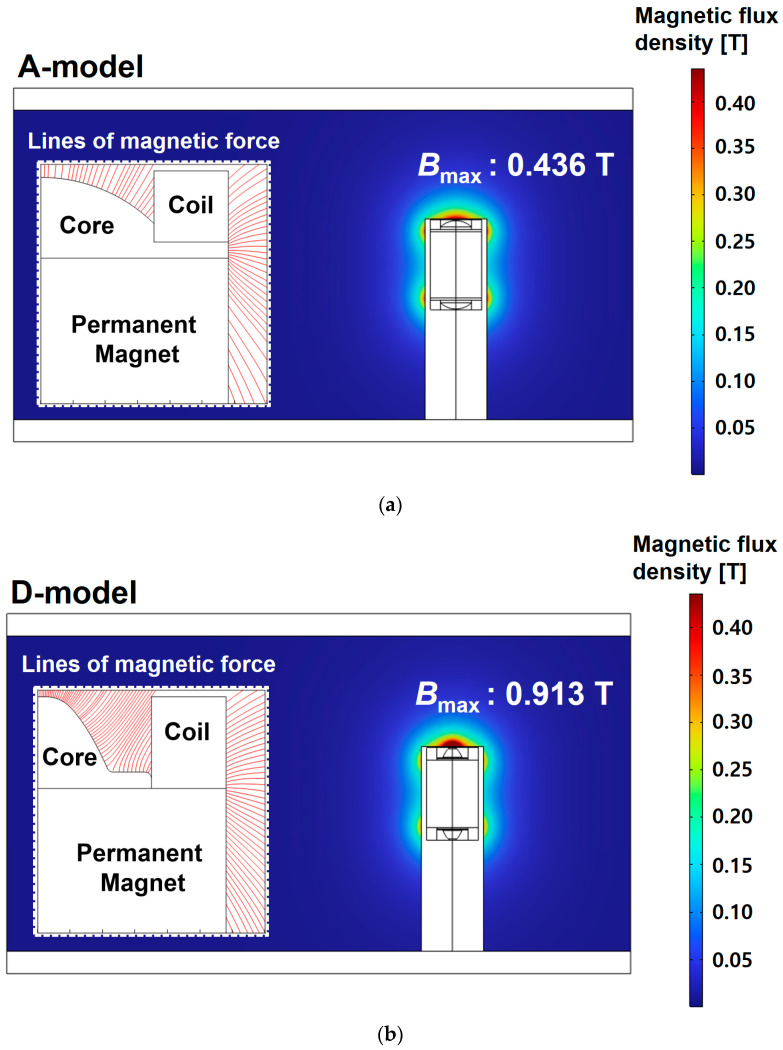
Magnetic flux density and magnetic force lines in the flow field: (**a**) A-model and (**b**) D-model.

**Table 1 sensors-23-05381-t001:** Geometries of the four sensor models.

Model	A-Model	B-Model	C-Model	D-Model
a [mm]	15	15	15	15
b [mm]	5.8	5.8	5.8	5.8
c [mm]	2.2	2.2	2.2	2.8
d [mm]	2.3	2.3	2.3	2.3
e [mm]	2.5	2.5	2.5	2.8
f [mm]	3.5	3.5	3.5	3.5
g [mm]	0.5	0.5	0	0
h [mm]	0.4	0.4	0.4	0.2
r_1_ [mm]	5	-	-	-

**Table 2 sensors-23-05381-t002:** Working conditions for numerical calculations.

Items	Condition	Items	Condition
Inlet (velocity)	0.02 m/s	Outlet	Outflow
Number of particles	15,000	Particle diameter	10 μm
Shape of the particles	Sphere	Material of particle	Steel
Particle relative Permeability	1000	Particle density	8030 kg/m^3^
Density of the lubricant	870 kg/m^3^	Absolute viscosity of the lubricant	0.04 Pa∙s

## Data Availability

Not applicable.

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
