# Peer review of "Numerical Approach and Verification Method for Improving the Sensitivity of Ferrous Particle Sensors with a Permanent Magnet"

_sensors, 2023, doi:10.3390/s23125381_

Round 1

Reviewer 1 Report

This study evaluated sensibility of an existing sensor using a multi-physics analysis method and improved it by changing the core shape inside the sensor. This study will contribute to the development of more effective ferrous particle sensors. The reviewer believed that the manuscript can be considered to be accepted after answer the following question:

1.     "ehance the ablity" should be followed by "of". (P1 18)

2.     Are the design parameters obtained by dismantling the sensor was verified in the experiment? (P3 86)

3.     The order of the previous table was Table 1, but here it is Table 4, missing Table 2 and Table 3. Please write in the correct order. (P7 207)

4.     This is straight from verse 2 to verse 5, missing the titles of verse 3 and 4. Please write in the correct format. (P12 295)

5.     The years listed in the references are either bolded or not bolded. Please check them again. (P13 379)

The English language of the article can be read normally and easily understood.

Author Response

Response to Reviewer 1

I would like to thank the reviewers for their thorough reviews and useful comments and insights. Changes suggested by the reviewers have been incorporated in the manuscript. I hope that the referees will find them satisfactory. If there are further changes to be made, I will be happy to comply. I prepared sincere responses to your comments as follows.

……………………………………………………………………………………………………………………….

This study evaluated sensibility of an existing sensor using a multi-physics analysis method and improved it by changing the core shape inside the sensor. This study will contribute to the development of more effective ferrous particle sensors. The reviewer believed that the manuscript can be considered to be accepted after answer the following question:

Point 1: "ehance the ablity" should be followed by "of". (P1 18)

Response 1: Thank you for informing the grammatical error. The contents of the abstract revised as follows.

This study aimed to improve sensitivity of ferrous particle sensors used in various mechanical systems such as engine to detect abnormalities by measuring the amount of ferrous wear particles generated by metal-to-metal contact. Existing sensors collect ferrous particles using a permanent magnet. However, its ability to detect abnormalities is limited because it only measures amount of ferrous particles collected on the top of the sensor. This study provided a design strategy to boost the sensitivity of an existing sensor using a multi-physics analysis method, and a practical numerical method was recommended to assess the sensitivity of the enhanced sensor. The sensor's maximum magnetic flux density was increased by around 210% compared to the original sensor by changing the core's form. In addition, in the numerical evaluation of the sensitivity of the sensor, the suggested sensor model has improved sensitivity. This study is important because it offered a numerical model and a verification technique that may be used to enhance the functionality of a ferrous particle sensor that uses a permanent magnet.

Point 2: Are the design parameters obtained by dismantling the sensor was verified in the experiment? (P3 86)

Response 2: Design parameters were measured through disassembly of existing products, and there may be slight size error. It was not tested through experiments. Since this study presents a design direction for increasing the sensitivity of this sensor and presents a numerical method, so a slight difference in dimensions is not considered important. I would appreciate it if you could understand that we did not consider the exact design values of existing sensors.

Point-3: The order of the previous table was Table 1, but here it is Table 4, missing Table 2 and Table 3. Please write in the correct order. (P7 207)

Response 3: I modified it as follows.

Table 4. ⇒ Table 2.

Point-4: This is straight from verse 2 to verse 5, missing the titles of verse 3 and 4. Please write in the correct format. (P12 295)

Response 4: I modified as follows.

  1. Conclusions ⇒ 3. Conclusions

Point-5: The years listed in the references are either bolded or not bolded. Please check them again. (P13 379)

Response 5:

I have corrected the year in bold. (Reference 4, 9)

I thank the reviewers again for their time and insights.

Sincerely yours,

Sung-Ho Hong

Reviewer 2 Report

1. The refinement part of the grid diagram in Figure 4 does not correspond to the simulation diagram.

2. The flux density obtained from the simulation in Figure 6 is only the flux density of the core of the head or the flux density of the whole magnet?

3. Iron particles are not charged can not apply Lorentz force calculation.

4. Where is the source of the cited formula for the trapping force of a particle in a fluid?

5. Where is the source of the quoted formula for the magnetic swimming force of iron particles?

6. The fluid simulation model is simple and not with the actual.

7. The fluid velocity is too small, and the results obtained have no reference value for practical applications.

Author Response

Response to Reviewer 2

I would like to thank the reviewers for their thorough reviews and useful comments and insights. Changes suggested by the reviewers have been incorporated in the manuscript. I hope that the referees will find them satisfactory. If there are further changes to be made, I will be happy to comply. I prepared sincere responses to your comments as follows.

……………………………………………………………………………………………………………………….

Point 1: The refinement part of the grid diagram in Figure 4 does not correspond to the simulation diagram

Response 1: I modified it to an appropriate picture and also modified the title of the picture. In addition, the information about the mesh for Figure 4 in the text is also modified.

Title: Figure 4. Meshes around sensor          

Contents: The total number of meshes was 158,525.

Point 2: The flux density obtained from the simulation in Figure 6 is only the flux density of the core of the head or the flux density of the whole magnet?

Response 2:  The flux density obtained from the simulation in Figure 6 is for the whole magnet.

So, I added the explanation about this.

Point-3: Iron particles are not charged can not apply Lorentz force calculation.

Response 3: The ferrous particles in this model are neutral particles, and magnetophoretic force is applied to the particles. The magnetophoretic force is expressed as Fext in equation (6). Factors that affect magnetophoretic force are particle radius, relative permeability, and magnetic field strength.

Point-4: Where is the source of the cited formula for the trapping force of a particle in a fluid?

Response 4: A function called “particle counter” of the particle tracing module was used, to check the number of particles attached to the sensor. The counter can detect all particles or only the particles released by a specified feature such as transmitted mass flow rate and transmitted current. Unfortunately, the related formula is not the program manual.

I added the explanation about this as follows.

To check the number of particles attached to the sensor, a function called “particle counter” of the particle tracing module was used.

Point-5: Where is the source of the quoted formula for the magnetic swimming force of iron particles?

Response 5: The magnetophoretic force acting on the ferrous particles is defined as Fext in equation (6).

I added the explanation in the text.

Point-6: The fluid simulation model is simple and not with the actual.

Response 6: As the reviewer commented, the actual gearbox or engine system is very complex. Numerical analysis about a real lubrication system may be appropriate. However, in terms of suggesting the direction of design to improve the sensitivity of the sensor and presenting a numerical model to verify it, the simple numerical model is considered sufficient. This is because the optimal sensor position considering the flow must be selected first, in order to verify the sensitivity of the sensor in the lubrication system where complex flow actually occurs.

So, I added the explanation about this in the text as follows.

Numerical analysis about a real lubrication system may be appropriate. However, the simple numerical model is deemed sufficient in terms of outlining a design path to in-crease the sensor's sensitivity and providing a numerical model to support it. This is due to the fact that the best sensor position for the flow must be chosen first in order to test the sensor's sensitivity in the lubricating system, which is where complex flow actually hap-pens. Therefore, a simple numerical model that did not generate complex flow was applied.

Point-7: The fluid velocity is too small, and the results obtained have no reference value for practical applications.

Response 7: As the reviewer commented, when hydraulic oil works, the flow rate is fast and the pressure is high. However, there are many places where low-velocity flows occur, such as by-passes connected to purification systems around sump tank. In this analysis, if the flow speed is very fast, it takes a lot of time to converge the analysis, so the very fast flow condition was not applied. It would be appreciated if you could understand that the analysis conditions were applied in terms of ease of numerical analysis.

I thank the reviewer again for their time and insights.

Sincerely yours,

Sung-Ho Hong

Reviewer 3 Report

This paper deals with the investigation of optimization sensitivity of ferrous particle sensors with permanent magnet. There are a lot of literature available in this field with manufacturing design, and hence the author could not bring any novelty in this manuscript. However, this work could be the basis for an interesting article. The author could polish up the work in terms of English.

Some of the queries need to be carefully before this paper can be considered for publication:

 1. Introduction is too long, author is suggested to pinpoint the novelty of this work.  The novelty and research gaps could be emphasized. First of all, a lot of work has already been published, why is your methodology unique and why have previous studies not addressed these details ?

2. In particular, there are only the numerical and analytical analysis in this work. The title is exaggerated to correlate this study in real case.  The research outputs did not justify the claims. PLease change the title

3. How does the proposed improved sensor design perform in scenarios where the ferrous particle concentration in the lubricating oil is extremely low or high, and how might the sensor's sensitivity be affected by such variations in particle concentration ?

4. How might the multiphysics analysis method employed in this study be further expanded to account for additional factors affecting the performance of the sensor, such as temperature or pressure?

5. What considerations should be made for designing sensors that can adapt to a wide range of fluid velocities “It is difficult to attach particles to the sensor under conditions where the velocity of the fluid is higher”?

6. In the proposed improved sensor design, how does the distribution of magnetic flux density affect the sensor's ability to differentiate between various sizes and types of ferrous particles?

7. How might the numerical analysis approach be extended to consider the interactions between ferrous particles and non-ferrous contaminants in the lubricating oil, and what impact could these interactions have on the sensor’s sensitivity and accuracy in detecting mechanical system abnormalities?

8.  Are there any potential trade-offs or drawbacks to the improved sensor design in terms of cost, manufacturability, or durability that may impact its widespread adoption?

9. What are the potential limitations of the multiphysics analysis method used in this study, and how might they affect the accuracy and applicability of the findings?

10. In what ways might the shape of the core inside the ferrous particle sensor be further optimized to enhance its sensitivity, and how might other design parameters, such as material properties and coil configurations, influence the performance of the sensor?

11. Can the methodology presented in this study be applied to improve the performance of other types of sensors, and what potential challenges might be faced in adapting the approach

12. The English language is clearly not proofread; there are syntax and spelling errors in every other paragraph. Sometimes these errors make it difficult to understand the meaning of the sentence.

  13.  At this time, I unfortunately cannot recommend this work. Recommend extending the details and  works to claim the credits in the conclusion made.

The English language is clearly not proofread; there are syntax and spelling errors in every other paragraph. Sometimes these errors make it difficult to understand the meaning of the sentence.

 Abstract: This study aimed to improve the sensitivity of ferrous particle sensors used in various mechanical systems such as wind power gearboxes to detect abnormalities anomalies by measuring the amount of ferrous wear particles generated by metal-to-metal contact. Existing sensors collect ferrous particles using a permanent magnet. However, its ability to detect abnormalities anomalies is limited because it only measures amounts of ferrous particles collected on the top of the sensor. To overcome this limitation, this study evaluated sensibility the sensitivity of an existing sensor using a multi-physics analysis method and improved it by changing the core shape inside the sensor. The detection ability of the improved sensor was analytically shown to be better than that of the existing sensor. This study highlights the use of analytical methods and multi-physics multiphysics analysis for developing the development of sensor for ferrous particles sensor and presents a method to compare sensibility analytically. Results It is expected that the results of the improved sensor are expected to will enhance the ability to detect abnormalities anomalies in the lubrication system where ferrous wear particles are generated. This study will contribute to the development of more effective ferrous particle sensors.

Author Response

Response to Reviewer 3

Dear Editor and Reviewer

I would like to thank the reviewers for their thorough reviews and useful comments and insights. Changes suggested by the reviewers have been incorporated in the manuscript. I hope that the referees will find them satisfactory. If there are further changes to be made, I will be happy to comply. I prepared sincere responses to your comments as follows.

……………………………………………………………………………………………………………………….

This paper deals with the investigation of optimization sensitivity of ferrous particle sensors with permanent magnet. There are a lot of literature available in this field with manufacturing design, and hence the author could not bring any novelty in this manuscript. However, this work could be the basis for an interesting article. The author could polish up the work in terms of English.

Some of the queries need to be carefully before this paper can be considered for publication:

Point 1: Introduction is too long, author is suggested to pinpoint the novelty of this work.  The novelty and research gaps could be emphasized. First of all, a lot of work has already been published, why is your methodology unique and why have previous studies not addressed these details?

Response 1: Some of the contents in the introduction were deleted and the contents were modified in terms of the novelty of this study as follows.

[deleted contents]

For over 50 years, the broadest definition of wear has been the loss of a material from a surface, the transfer of a material from one surface to another, or the movement of a material within a single surface [5]. A narrower definition of wear has been suggested as progressive loss of a substance from the operating surface of a body occurring as a result of relative motion at the surface [6]. It is more beneficial for tribologists to utilize a wider definition, given the range of engineering applications involved. A simple and effective definition of wear is damage to a solid surface caused by progressive material loss, usually due to relative motion between that surface and a substance or substances in contact with it [7,8].

[revised contents]

While there have been many studies on the sensitivity of wearable particle sensors during development, there have been relatively few studies on the sensitivity of ferrous particle sensors [31-36]. Improvements in the sensor's signal processing, modifications to its internal design, and other techniques may be used to increase the sensitivity of the ferrous particle sensor. The ferrous particle sensor has cylindrical permanent magnet shape inside. As a result, ferrous particles are gathered not only on the sensor's top but also on its sides. So, it is difficult to reflect the exact change in the amount of ferrous particles attached to the sensor. This study focused on improving the sensitivity of the sensor by changing the design inside the sensor. This procedure involved creating a numerical model for the sensor in order to change its design effectively, and presenting a numerical verification model to assess how sensitive the modification was. In addition, a new design model that improves the sensitivity of the existing sensor was presented, and the improvement was shown through an effective numerical verification model in terms of cost and time.

Point 2: In particular, there are only the numerical and analytical analysis in this work. The title is exaggerated to correlate this study in real case.  The research outputs did not justify the claims. Please change the title

Response 2: I changed the title as follows.

(Before) Title: Improving Sensitivity of Ferrous Particle Sensors with Permanent Magnet

(After) Title: Numerical Approach and Verification Method for Improving Sensitivity of Ferrous Particle Sensors with Permanent Magnet

Point-3: How does the proposed improved sensor design perform in scenarios where the ferrous particle concentration in the lubricating oil is extremely low or high, and how might the sensor's sensitivity be affected by such variations in particle concentration?

Response 3: What we used for analysis was performed by uniformly injecting 15,000 particles in the particle injection area. However, if the number of particles is reduced, the rate of collection may be reduced. The reason is that the distribution of magnetic flux density formed around the sensor is constant, but if the total number of particles decreases, the number of particles going around the sensor also decreases. In addition, if the particles are non-uniformly injected in the particle injection area, the rate of collection will be different.

So, I added an explanation about this as follows.

In addition, the ratio of particles collected by the sensor will change according to the number of injected particles. The reason is that the magnetic flux density of the sensor is constant and the number of particles directed around the sensor decreases when the number of injected particles decreases.

Point-4: How might the multiphysics analysis method employed in this study be further expanded to account for additional factors affecting the performance of the sensor, such as temperature or pressure?

Response 4: In this analysis, only the velocity of the fluid was changed, but if temperature or pressure is used as the boundary condition, analysis according to the change in temperature or pressure conditions is possible If the change in temperature is to be taken into account, the energy equation is added as the governing equation.

So, I added an explanation about this as follows.

Moreover, when pressure or temperature is a major factor in the analysis conditions, sufficient analysis is possible as a boundary condition. To take into consideration the change in temperature, the energy equation is additionally employed in the governing equations.

Point-5: What considerations should be made for designing sensors that can adapt to a wide range of fluid velocities “It is difficult to attach particles to the sensor under conditions where the velocity of the fluid is higher”?

Response 5: In particular, when the flow rate increases, the inertial force of the particles in the flow also increases, so it is difficult to diagnose the condition with a single sensor. Therefore, research on selecting an appropriate number of sensors and arranging them in an appropriate location should be additionally performed. In the case of small flow velocities, it is desirable to place the sensor where the main flow forms.

Point-6: In the proposed improved sensor design, how does the distribution of magnetic flux density affect the sensor's ability to differentiate between various sizes and types of ferrous particles?

Response 6: In fact, this sensor has a function to distinguish large particles from small particles, which can be known by the change in inductance formed around the coil. However, this sensor cannot distinguish the shape of iron wear particles. In general, the shape of the particle is distinguished using an optical sensor. In this study, the sensitivity of the sensor was evaluated by the number of ferrous particles attached to the sensor. Unfortunately, the distribution of magnetic flux density in the proposed sensor design does not affect the ability of the sensor to discriminate ferrous particles of various sizes and types.

Point-7: How might the numerical analysis approach be extended to consider the interactions between ferrous particles and non-ferrous contaminants in the lubricating oil, and what impact could these interactions have on the sensor’s sensitivity and accuracy in detecting mechanical system abnormalities?

Response 7: In an actual lubrication system, there are not only magnetic particles but also non-magnetic particles. So the interaction between the two particles also appears. However, since this sensor is applied only to detect magnetic particles, it is impossible to detect non-magnetic particles and it is difficult to consider the interaction between the two particles.

Point-8: Are there any potential trade-offs or drawbacks to the improved sensor design in terms of cost, manufacturability, or durability that may impact its widespread adoption?

Response 8: Since this sensor measures the amount of ferrous particles after attaching iron particles to the sensor, it is important that the sensor collects the ferrous particles in the lubricating oil. Therefore, in terms of broad usability, research on optimal positioning of the sensor should also be conducted.

So, I added an additional explanation as follows.

The optimal sensor position considering the flow must be selected first, in order to verify the sensitivity of the sensor in the lubrication system where complex flow actually occurs.

Point-9: What are the potential limitations of the multiphysics analysis method used in this study, and how might they affect the accuracy and applicability of the findings?

Response 9: This analysis is approached using multi-physics analysis, but it is difficult to consider all realistic conditions. In particular, the number of particles can be less than 1% of the total fluid volume, and the shape of the particles can be interpreted only as a sphere. Actual wear particles include non-magnetic particles and also consist of very diverse types of particles. However, in terms of approximate evaluation, a suitable numerical analysis is more useful in terms of cost and time than experimental analysis.

So, I added an additional explanation about limitation of this numerical method as follows.

Actual wear particles include non-magnetic particles and also consist of very diverse types of particles. This analysis is limited by the fact that the total number of particles can be less than 1% and that the shape of the particles can only be described as spherical.

Point-10: In what ways might the shape of the core inside the ferrous particle sensor be further optimized to enhance its sensitivity, and how might other design parameters, such as material properties and coil configurations, influence the performance of the sensor?

Response 10: If the material of the core is changed, the B-H curve and relative magnetic permeability change. However, in this study, increasing of the maximum magnetic flux density by changing the shape of the core inside the sensor is focused because the measurement is effective only when the particles are well collected on the upper part of the sensor. In other words, since it was determined that concentrating the magnetic flux density on the top of the sensor would improve the sensitivity of the sensor, the shape of the core was concentrated on the top to concentrate the magnetic flux density on the top. In this process, the reluctance in the side direction is smaller than the top direction, so a lot of magnetic flux escapes.

Point-11: Can the methodology presented in this study be applied to improve the performance of other types of sensors, and what potential challenges might be faced in adapting the approach

Response 11: The presented research method is considered to be useful for sensor development if electromagnetic analysis and flow analysis are used for the development of sensors using inductance and capacitance among wear sensors. However, when developing a sensor, since the aspects of efficiency, usability, manufacturing and stability will also be considered, these parts should be considered as constraints in the development process of the sensors.

Point-12: The English language is clearly not proofread; there are syntax and spelling errors in every other paragraph. Sometimes these errors make it difficult to understand the meaning of the sentence.

Response 12: This revised manuscript received English proofreading from a professional institution. So spelling errors and grammar have been improved.

Point-13: At this time, I unfortunately cannot recommend this work. Recommend extending the details and works to claim the credits in the conclusion made.

Response 13: I added explanations for the comments made by the reviewers and corrected the parts that required correction. I would be grateful if you could review the revised manuscript.

he English language is clearly not proofread; there are syntax and spelling errors in every other paragraph. Sometimes these errors make it difficult to understand the meaning of the sentence.

Abstract: This study aimed to improve the sensitivity of ferrous particle sensors used in various mechanical systems such as wind power gearboxes to detect abnormalities anomalies by measuring the amount of ferrous wear particles generated by metal-to-metal contact. Existing sensors collect ferrous particles using a permanent magnet. However, its ability to detect abnormalities anomalies is limited because it only measures amounts of ferrous particles collected on the top of the sensor. To overcome this limitation, this study evaluated sensibility the sensitivity of an existing sensor using a multi-physics analysis method and improved it by changing the core shape inside the sensor. The detection ability of the improved sensor was analytically shown to be better than that of the existing sensor. This study highlights the use of analytical methods and multi-physics multiphysics analysis for developing the development of sensor for ferrous particles sensor and presents a method to compare sensibility analytically. Results It is expected that the results of the improved sensor are expected to will enhance the ability to detect abnormalities anomalies in the lubrication system where ferrous wear particles are generated. This study will contribute to the development of more effective ferrous particle sensors.

Response: The contents of the abstract revised as follows.

This study aimed to improve sensitivity of ferrous particle sensors used in various mechanical systems such as engine to detect abnormalities by measuring the amount of ferrous wear particles generated by metal-to-metal contact. Existing sensors collect ferrous particles using a permanent magnet. However, its ability to detect abnormalities is limited because it only measures amount of ferrous particles collected on the top of the sensor. This study provided a design strategy to boost the sensitivity of an existing sensor using a multi-physics analysis method, and a practical numerical method was recommended to assess the sensitivity of the enhanced sensor. The sensor's maximum magnetic flux density was increased by around 210% compared to the original sensor by changing the core's form. In addition, in the numerical evaluation of the sensitivity of the sensor, the suggested sensor model has improved sensitivity. This study is important because it offered a numerical model and a verification technique that may be used to enhance the functionality of a ferrous particle sensor that uses a permanent magnet.

I thank the reviewer again for their time and insights.

Sincerely yours,

Sung-Ho Hong

Round 2

Reviewer 2 Report

1. There is no cited source for the tracing force formula.

2. There is no cited source of magnetophoretic force equation.

3. The fluid simulation model is simple and does not match with the reality.

4. The fluid velocity is too small, and the results obtained have no reference value for practical applications.

5. Insufficient innovation points.

Author Response

Dear Editor and Reviewer

I prepared sincere responses to your comments as follows.

Point 1: There is no cited source for the tracing force formula.

Answer 1: I added a reference as follows.

where mp, v1, FD, and Fext are particle mass [kg], velocity vector of the particle [m/s], drag force [N], and magnetophoretic force [N], respectively. In addition, τp, dp, rp, ρp, μ, μ0, μr, and K mean particle velocity response time [s], particle diameter [m], particle radius [m], parti-cle density [kg/m3], viscosity of fluid [Pa∙s], vacuum permeability [kg∙m∙s/A2], relative per-meability, and nondimentional parameter, respectively [38].

Point 2:  There is no cited source of magnetophoretic force equation.

Answer 2: I added a reference as follows.

Magnetophoretic force is expressed as Fext in equation (6)

                    (6)

where mp, v1, FD, and Fext are particle mass [kg], velocity vector of the particle [m/s], drag force [N], and magnetophoretic force [N], respectively. In addition, τp, dp, rp, ρp, μ, μ0, μr, and K mean particle velocity response time [s], particle diameter [m], particle radius [m], parti-cle density [kg/m3], viscosity of fluid [Pa∙s], vacuum permeability [kg∙m∙s/A2], relative per-meability, and nondimentional parameter, respectively [38].

Point 3:  The fluid simulation model is simple and does not match with the reality.

Answer 3: I partially agree with the reviewer’s comment. I selected an appropriate analysis model to numerically analyze the improvement of the particle collection ability of the ferrous particle sensor. However, I believe that this paper is novel in that it suggests a numerical method using multi-physics that can numerically evaluate the ferrous particle sensor and its sensitivity. Although it must be verified experimentally through testing, numerical method is more effective in terms of time and cost than the experimental method, and is of great help in determining the direction of design improvement. In that respect, although it is not an analysis of actual mechanical system, it is meaningful in that it presents an appropriate numerical model and evaluation method about sensitivity.

So, I revised an explanation as follows.

This numerical analysis model was proposed simply and uncomplicated to numerically evaluate the sensitivity of the improved ferrous particle sensor, not to analyze a specific mechanical system.

Point 4:  The fluid velocity is too small, and the results obtained have no reference value for practical applications.

Answer 4: I agree with the reviewer’s comment. In the case of hydraulic oil, it corresponds to a case where the flow rate is very fast. If the analysis is performed for the case where the flow rate is high, the number of particles attached to the ferrous particle sensor will be very small due to the inertial force of the fluid. In addition, when the flow rate is high, the position of the sensor will have a greater effect on the sensitivity of the sensor, and additional improvements such as installing a flow guide wall around the sensor should be provided. In relation to this part, a paper on the optimal positioning of the ferrous particle sensor in the gearbox is currently under review in another journal (sensors and actuators).

So, the explanations related to the reviewer’s comment, were added as follows.

The velocity of the flow in the analysis is small, so it is different from the actual hydraulic fluid. When the velocity of the fluid is high, the number of particles attached to the ferrous particle sensor decreases due to the inertial force of the fluid. In addition, when the speed of the fluid increases, the influence of the positioning of the sensor increases. Therefore, in order to easily compare the sensitivity improvement of the sensor, an analysis was performed under the condition of low speed.

Point 5: Insufficient innovation points.

Answer 5: As the reviewer's comments, this study is not general because it is not an analysis of the actual mechanical system, and the velocity condition of the fluid is also very small. However, in the condition diagnosis using the ferrous particle sensor, there were many cases in which the sensor did not measure even though abnormal wear occurred. In order to solve this problem, I think that optimal positioning should be done along with sensitivity improvement through changing the internal design of the sensor. As mentioned above, this paper is novel in that it first proposed a numerical analysis approach for ferrous particle sensors and presented effective numerical verification method in terms of time and cost. Moreover, I suggested new design about core shape to improve sensitivity of the sensor.

It is thought that the completeness of our paper has been improved more through the correction of each comment.

Thank you once again for reviewing our paper.

Sincerely yours,

Sung-Ho Hong

Reviewer 3 Report

Accepted

Author Response

I appeciate your review.